# Tetrodotoxin and Its Analogues in *Cephalothrix* cf. *simula* (Nemertea: Palaeonemertea) from the Sea of Japan (Peter the Great Gulf): Intrabody Distribution and Secretions

**DOI:** 10.3390/toxins12120745

**Published:** 2020-11-26

**Authors:** Anna E. Vlasenko, Timur Yu. Magarlamov

**Affiliations:** A.V. Zhirmunsky National Scientific Center of Marine Biology, Far Eastern Branch, Russian Academy of Sciences, 690041 Vladivostok, Russia; avlasenko@imb.dvo.ru

**Keywords:** tetrodotoxin, tetrodotoxin analogues, TTX, TTX distribution, Nemertea, *Cephalothrix*, HPLC–MS/MS

## Abstract

Some nemertean species from the genus *Cephalothrix* accumulate tetrodotoxin (TTX) in extremely high concentrations. The current study is the first to provide high-performance liquid chromatography–tandem mass spectrometry (HPLC–MS/MS) data on tetrodotoxin and its analogues (TTXs) profile and concentration in different regions and organs of *Cephalothrix* cf. *simula*, and its secretions produced in response to stimulation. Different specimens of *C*. cf. *simula* possessed 7–11 analogues, including nine previously found in this species and two new for nemerteans—4,9-anhydro-8-epi-5,6,11-trideoxyTTX and 1-hydroxy-8-epi-5,6,11-trideoxyTTX. The study of the toxins’ distribution in different regions and organs of nemerteans revealed the same qualitative composition of TTXs throughout the body but differences in the total concentration of the toxins. The total concentration of TTXs was highest in the anterior region of the body and decreased towards the posterior; the ratio of the analogues also differed between regions. The data obtained suggest a pathway of TTXs uptake in *C.* cf. *simula* and the role of toxins in the life activity of nemerteans.

## 1. Introduction

Tetrodotoxin (TTX) is a potent low-molecular-weight neurotoxin of bacterial origin found in many types of animals and dinoflagellates. It is assumed that carnivores, omnivores, and scavengers can obtain TTX via the food chain or through parasitism and symbiosis [1]. Ribbon worms (Nemertea) are one of these animals that can accumulate TTX in high amounts. Nemertea is a small group (more than 1250 species) of bilateral, coelomic, unsegmented marine worms, most of which are active predators [2]. TTX is mosaically distributed in all three phylogenetic groups of Nemertea—palaeonemertea, pilidiophora, and hoplonemertea. Among palaeonemertea, TTX is found in the genera *Cephalothrix* [3,4,5,6,7,8] and *Tubulanus* [8,9], among pilidiophora, in species of the genera *Ramphogordius*, *Riseriellus* [4], *Dushia*, *Micrura* [10], *Lineus* [4,7,9], *Cerebratulus*, and *Kulikovia* [8]; among hoplonemertea, in representatives of the *Amphiporus* [4,7] and *Nipponnemertes* [7].

Of particular interest are nemerteans of the genus *Cephalothrix*, which can accumulate extremely high amounts of TTX, which puts them at the same level of toxicity as puffer fishes and blue-ringed octopuses [7,8,11]. To date, there are several studies on mice showing a strong toxic TTX-associated effect of *Cephalothrix simula* extracts [5,6,7]. In subsequent studies, TTX and a number of its analogues were identified in this species [8,11]. The distribution of TTX-like substances in tissues of *Cephalothrix* sp. was also visualized using an immunohistological study [12]. According to a generally accepted opinion, animals use the toxin for defense, and some species also use it as a weapon [13,14]. Ali and colleagues [3] revealed TTX in the proboscis and secretions of *C. simula* (named *Cephalothrix linearis*) released by nemerteans in response to stimulation, which may indicate the use of the toxin both to repel predators and capture prey. However, comprehensive studies showing the distribution of TTX and its analogues (TTXs) in highly toxic nemerteans of the genus *Cephalothrix,* necessary to understand the transfer and functions of the toxins in animals, have not been performed. In the current work, TTXs were analyzed in different regions and organs of *C.* cf. *simula*, as well as in secretions released by nemerteans upon stimulation. The profile and concentration of TTX and its analogues in the intestine, body wall, proboscis, and secretions were obtained using high-performance liquid chromatography with tandem mass spectrometry (HPLC–MS/MS), suggesting possible pathways of toxin uptake and transfer in the body of nemerteans and their functions.

## 2. Results

TTXs distribution in the nemertean *Cephalothrix* cf. *simula* was investigated in two experiments using HPLC–MS/MS. The first experiment studied the TTXs’ profile of the secretions and body of *C.* cf. *simula* after stimulation. During the second experiment, TTXs’ distributions in the proboscis, different regions of the intestine, and the body wall of *C.* cf. *simula* were studied.

As a result of the first experiment, seven TTXs were detected in extracts of the body and secretions: TTX, 4-epiTTX, 11-norTTX-6(s)-ol, 4,9-anhydroTTX, 11-deoxyTTX, 5-deoxyTTX, and 5,6,11-trideoxyTTX. As a result of the second experiment, 11 toxins—the seven previously listed and 11-norTTX-6(r)-ol, 11-oxoTTX, 4,9-anhydro-8-epi-5,6,11-trideoxyTTX, and 1-hydroxy-8-epi-5,6,11-trideoxyTTX—were additionally identified (Figure 1, Table 1).

The TTXs’ amount in *C.* cf. *simula* individuals in both experiments varied from the minimum concentration of 0.13 mg TTXs per gram of tissue (specimen No. 7) to the maximum of 2.14 mg TTXs per gram of tissue (specimen No. 1) (Table 1 and Table 2).

### 2.1. TTXs Secretion by Cephalothrix cf. simula

The percentage of toxins excreted by animals during stimulation with an electric current compared to the total concentration of body and secretions toxins varied from 0.2% in specimen No. 1 to 1.4% in specimen No. 3 (Table 1).

In extracts from both the body and the secretions of specimens No. 1 and No. 3, the predominant toxins were TTX and 5,6,11-trideoxyTTX. They comprised more than 72% of the total amount of TTXs. At the same time, in extracts from No. 2 and No. 4, a significant contribution was made by the third analogue, 5-deoxyTTX, giving more than 82% of all TTXs in total with TTX and 5,6,11-trideoxyTTX. In body extracts of individuals No. 2 and No. 3, the predominant toxin was 5,6,11-trideoxyTTX at 66.1% and 59.2%, respectively. In extracts of the body and secretions of specimens No. 1 and No. 4, TTX was the predominant toxin at 46.2% and 70.5%, respectively.

The percentage of TTX and its analogues varied both in the body and secretions. Thus, the content of TTX in the body ranged from 26.7% to 59.6%, and in the secretions it reached 46.2%–70.5%. The contents of some TTXs—4-epiTTX, 4,9-anhydroTTX, and 5-deoxyTTX—in the extracts also differed and were relatively higher in secretions compared to those in the body. The content of 4-epiTTX in the body extracts ranged from 1.0% to 2.5%, and the content in the secretions ranged from 3.3% to 11.6%; the proportion of 4,9-anhydroTTX ranged from 0.05% to 0.47% in the body, and the proportion in the secretions ranged from 0.65% to 6.60%; the proportion of 5-deoxyTTX ranged from 0.8% to 2.9% in the body and from 9.6% to 22.3% in secretions. The content of 5,6,11-trideoxyTTX was lower in secretions compared to the body: 28.8%–66.1% in the body versus 13.8%–26.1% in the secretions (Figure 2).

### 2.2. Distribution of TTXs in the Body of Cephalothrix cf. simula

The major toxins of the intestine, body wall, and proboscis samples of three *C.* cf. *simula* specimens were TTX, 5-deoxyTTX and 5,6,11-trideoxyTTX, which together accounted for 93.9%–98.6% of all TTX analogues. The minor toxins were 4-epiTTX, 11-norTTX-6 (S)-ol, 11-norTTX-6 (R)-ol, 4,9-anhydroTTX, 11-deoxyTTX, 11-oxoTTX, 4,9-anhydro-8-epi-5,6,11-trideoxyTTX, and 1-hydroxy-8-epi-5,6,11-trideoxyTTX; the minor toxins’ combined amount accounted for 1.4%–6.1% (Table 2).

The absolute amount of TTXs in different organs varied greatly and differed several-fold between some specimens. Thus, the ratios of TTXs in the intestine, body wall, and proboscis for specimens No. 5, No. 6, and No. 7 were 15:2:5, 6:13:2, and 17:11:2, respectively. It can be seen from the ratio that the lowest concentrations of TTXs were found in the proboscises of specimens No. 6 and No. 7, and the body wall of specimen No. 1. The highest concentrations of TTXs were found in the intestines of specimens No. 1 and No. 3 and in the body wall of specimen No. 2 (Figure 3).

TTXs analysis in the body of nemerteans (body wall + intestine) showed that body toxins in specimens No. 5, No. 6, and No. 7 were located mainly in the anterior region: 80.4%, 59.0%, and 71.8%, respectively. The remaining body toxins in specimens No. 5, No. 6, and No. 7 were distributed between the middle and posterior regions; the contents of TTXs in the middle region were 1.25, 1.8, and 3.4 times higher, respectively, than those in the posterior region (Table 2).

Distributions of major TTXs in the anterior, middle, and posterior regions of *C.* cf. *simula* body were different for each analogue. The largest amount of TTX was in the anterior region in specimens No. 5, No. 6, and No. 7–56.7%, 67.4%, and 65.6%, respectively, while percentages in the posterior region decreased by 1.4–1.7 times and amounted to 35.9%, 46.7%, and 38.9%, respectively. 5-deoxyTTX was mostly widespread in the posterior region of all specimens, where its content was 2–2.1 times higher than that in the anterior region. The content of 5,6,11-trideoxyTTX was practically equal in the middle and posterior regions of specimens No. 6 and No. 7 and was 1.4–1.5 times higher than that in the anterior region of the same specimen. In specimen No. 5, the content of this analogue in all sections was approximately the same (Table 2).

For minor TTXs, no large differences were found in the contents for different regions; however, in the posterior region of nemerteans, the amounts of some toxins—4-epiTTX, 4,9-anhydroTTX, and 1-hydroxy-8-epi-5,6,11-trideoxyTTX—were higher compared to that in the anterior and middle regions; conversely, the proportion of 11-norTTX-6 (S)-ol was less (Table 2).

The contribution of proboscis toxins to the total toxins of the organism varied greatly between different specimens and amounted to 21.9%, 9.4%, and 6.7% of total TTXs for specimens No. 5, No. 6 and No. 7, respectively. In the proboscis of all three specimens, major toxins were present at the same levels as in the body; the sum of TTX, 5-deoxyTTX, and 5,6,11-trideoxyTTX accounted for 89.7%–98.6%. The most represented toxin was TTX—38.9%–63.2% of all TTXs. Among the minor analogues, 11-norTTX-6 (S)-ol can be distinguished; in all specimens, its amount in the proboscis was higher than in each region of the intestine and body wall. In the proboscis of specimen No. 5, 11-oxoTTX was found at an order of magnitude higher than in the body (Table 2).

## 3. Discussion

An extremely high amount of toxins was shown for nemertean species from the genus *Cephalothrix*. The content of TTXs in individuals of *Cephalothrix simula* (named *Cephalothrix linearis*) collected from the intertidal zone in Shimoda (Shizuoka prefecture, Japan) reached 23,000 MU/g [3], and the content of TTXs in *Cephalothrix simula* individuals caught in the Hiroshima Bay (Japan) reached 25,590 MU/g [6,7]; these are equivalent to the concentrations of 4094.0–4555.0 μg/g TTX, respectively. According to Asakawa et al. [7], approximately 80% of *C. simula* collected in Hiroshima Bay in 1998–2005 were “extremely toxic”—their toxicity was higher than 1000 MU/g (178 μg/g) (according to the toxicity classification established by the Ministry of Health, Labor and Welfare of Japan). In the current work, *C.* cf. *simula*, belonging to the group of species *Cephalothrix simula* s. l, was investigated [15]. The concentration of TTXs in nemerteans varied from 133.3 (No. 7) to 2143 μg/g body (No. 1), while five out of seven examined individuals could be classified as “extremely toxic”, as their TTXs concentration was above 178 μg/g. Asakawa et al. [7] reported that species of the genus *Cephalothrix* from Qingdao (China) were less toxic than Japanese species. Combined samples of *C. simula* (=*C.* cf. *simula*) individuals collected from the Peter the Great Gulf (the Sea of Japan) earlier had a total concentration of TTXs at 66.8 [8] and 23.6 μg/g [16]. An individual of *C. simula* collected for the first time along the England coast had a total TTXs concentration of 54.3 μg/g [11]. The high toxicity of nemerteans from the coastal waters of Japan is likely associated with the ecological characteristics of the habitat needed to be clarified.

In some TTX-bearing nemertean species, along with TTX, analogues were also found. In most of the species, including *Amphiporus* sp., *Malacobdella japonica*, *Nipponnemertes punctatula*, *Lineus alborostratus* (=*Kulikovia alborostrata*), *Lineus fuscoviridis*, and *Lineus torquatus*, only equilibrium TTX analogues—4-epiTTX and 4,9-anhydroTTX—were found [17]. However, nonequilibrium analogues were also found in some nemertean species: 5-deoxyTTX, 11-deoxyTTX, and 5,6,11-trideoxyTTX were detected in *Kulikovia manchenkoi* [8] and 11-norTTX-6 (S)-ol and 5,11-dideoxyTTX were detected in *Yininemertes pratensis* [18]. Previously in nemerteans from the *Cephalothrix simula* s. l group, the nonequilibrium analogues 5-deoxyTTX, 11-deoxyTTX, 5,6,11-trideoxyTTX [16], 11-norTTX-6(S)-ol, 11-norTTX-6(R)-ol [8], 6,11-dideoxyTTX, 11-oxoTTX, 4,9-anhydro-5,6,11-trideoxyTTX, and 4,9-anhydro-11-oxoTTX [11] were previously found, along with TTX and its equilibrium analogues. Thus, in addition to TTX, 12 analogues belonging to all four families of TTX analogues were found in nemerteans: 1) equilibrium TTX analogues—4-epiTTX and 4,9-anhydroTTX; 2) deoxy analogues—5-deoxyTTX, 11-deoxyTTX, 5,6,11-trideoxyTTX, 6,11-dideoxyTTX, 5,11-dideoxyTTX, 4,9-anhydro-5,6,11-trideoxyTTX, and 4,9-anhydro-11-oxoTTX; 3) analogues with a removed methylene group at C-11—11-norTTX-6 (S)-ol and 11-norTTX-6 (R)-ol; 4) 11-CH_2_OH oxidized analogue—11-oxoTTX.

In the present study, different individuals of *C.* cf. *simula* were shown to possess 7–11 analogues, including nine already found in this species and two TTX analogues that were new for nemerteans—4,9-anhydro-8-epi-5,6,11-trideoxyTTX and 1-hydroxy-8-epi-5,6,11-trideoxyTTX. For three years, our research group has been studying the qualitative and quantitative composition of TTXs of the nemertean *C. simula* (=*C.* cf. *simula*) from Spokoinaya Bay, Peter the Great Gulf, Sea of Japan. In the work of 2018, more than 94% of all TTXs in the extract of *C*. *simula* (=*C*. cf. *simula*) comprised TTX, 5-deoxyTTX, and 5,6,11-trideoxyTTX, with a predominance of 5,6,11-trideoxyTTX (39.9%) [8]. In the nemertean caught in 2019, the proportion of toxins differed: the most represented TTX analogue was 4-epiTTX (46.0%), followed by TTX (31.8%) and 5,6,11-trideoxyTTX (20.1%) [16]. In nemerteans studied in the current work, TTX, 5-deoxyTTX, and 5,6,11-trideoxyTTX accounted for, in total, 93.9%–98.6% of all TTX analogues. 5,6,11-trideoxyTTX was the predominant toxin in individuals No. 2 and No. 3 (66.1% and 59.2%); TTX prevailed in the remaining nemerteans (46.2%–70.5%). In the specimen of *C. simula* studied by Turner and colleagues [11], the amount of TTX was 64%, the second was 6,11-dideoxyTTX (21%), which was not found in the current study, and the sum of 5-deoxyTTX (11-deoxyTTX) and 5,6,11-trideoxyTTX accounted for 9.9%. At the same time, 11-oxoTTX accounted for 5%, while in the current study, this analogue exceeded the limit of quantification (LoQ) only in the proboscis of specimen No. 5, and its amount was 0.03% of the total TTXs of the animal. Data obtained for *Cephalothrix simula* s. l. species do not show the obvious regularities in the qualitative and quantitative changes in the composition of TTX and its analogues. A correlation between the qualitative composition of toxins in foods and the composition of toxins in animals has been shown in several works. For example, feeding octopuses *Octopus vulgaris* with mollusks containing paralytic shellfish toxins, a group of low-molecular-weight guanidine toxins, resulted in the same prevalence of these toxins in both predator and prey [19]. However, in experiments studying the consumption of mollusks containing TTX (without analogues) by a puffer fish, *Takifugu obscurus*, TTX analogues were detected in the body of the fish, which might indicate possible biotransformation of toxins [20]. In the case of the nemerteans *C.* cf. *simula*, identical qualitative compositions of toxins in the intestine and body wall may indicate that TTX analogues appear due to food uptake, not as a result of TTX conversion. To test this assumption, *C*. cf. *simula* food item identification with subsequent analysis of TTX and its analogues are planned in the future.

According to the prevailing theory, TTX is produced by bacteria and transfers to animals capable of accumulating it. Animals at high trophic levels can accumulate TTX in high amounts via the food chain [1]. Thus, numerous feeding experiments on puffer fish [20], gill slug *Pleurobranchaea maculata* [21], and sea gastropod *Neptunea arthritica* [22] with TTX-containing items have shown the ability of animals to accumulate TTX from food. TTX uptake by nemerteans can also be a result of absorbing the toxin along with food items. Tanu et al. [12] investigated TTX localization in the body of nemertean *Cephalothrix* sp. by immunohistochemistry and showed that the only TTX-positive structures in the digestive tract were phagosomes of ciliated or “columnar” (according to Gibson, [23]) cells. The authors concluded that the main mechanism of TTX uptake into the nemertean body is the absorption of toxins along with the victim’s tissues through the phagocytic activity of ciliary cells. We confirmed by HPLC–MS/MS that the intestines of the studied nemerteans contained high amounts of TTXs in relation to the total content of TTX and its analogues in a nemertean’s body. We should also mention that the anterior intestine contained up to 85% of all intestinal toxins (Table 2). The inhomogeneous distribution of toxins in the digestive tract of *C.* cf. *simula* can result from different aspects in the digestion process. The process of food digestion in nemerteans with a macrophage type of nutrition, which include the genus *Cephalothrix*, can be divided into two phases of intracellular digestion which occur simultaneously in real conditions [23]. The first phase takes place in the foregut cavity, where the tissue of the victim is destroyed by proteolysis in an acidic environment (pH is approximately 5.0). The second phase involves intracellular digestion of small particles by ciliary cells located throughout the digestive tract. Thus, a high concentration of TTXs in the anterior region of the digestive tract of *C.* cf. *simula* can arise due to the relatively high local concentration of substances released from the food object during the destruction of its tissues in the foregut cavity, as well as due to the action of an acidic environment that increases TTXs solubility. Another issue is the inhomogeneous distribution of TTX analogues in different regions of the nemertean body. We found that while the content of TTX was highest in the anterior region in comparison to the middle and posterior regions, the amount of 5-deoxyTTX was the highest in the posterior region of all specimens. The content of 5,6,11-trideoxyTTX in the middle and posterior regions of two of the three nemerteans was higher than that in the anterior. It remains unclear whether such specific analogue localization is associated with different toxin absorptions due to differences in physicochemical properties or to the possible existence of an additional TTXs transfer mechanism, selectively acting towards certain analogues. Confirmation of a specific TTX transport mechanism was shown in toxic and conditionally toxic puffer fish in an in vitro experiment which incubated fish liver tissue slices in a toxin solution [24,25]. It was supposed that TTX was transported across the cell membrane of puffer fish liver cells using a carrier-mediated transport system [24].

TTX-bearing animals can use the toxin for defense against predators, for capturing and immobilizing the victim during hunting, and as a carrier of information according to some evidence [13]. Animals using TTX in venom accumulate the toxin in the organs involved in food capture and use it to immobilize the prey. In the blue-ringed octopuses, *Hapalochlaena lunulata* and *Hapalochlaena maculosa*, the bulk of TTX was found in the posterior salivary glands, a usual tool for producing paralyzing poison [26,27]. In the polyclad flatworm *Planocerid* sp., a large amount of TTX and 11-norTTX-6(S)-ol is contained in the pharynx and is utilized in the process of hunting [28]. In sea arrowworms, *Parasagilta elegans* (type Chaetognatha), TTX and its analogues were mainly localized in the head and, according to the authors’ assumptions, in the organs responsible for food capture [29]. Ali et al. [3] revealed that high amounts of TTX and its analogues were localized in the proboscis of nemertean *C. simula* (named *C. linearis*), a specific organ involved in the process of prey capture. When the proboscis is everted, cells of the glandular epithelium, according to immunohistochemical studies containing TTX-positive tags, appear outside [12,30]. It is known that nemerteans with macrophage-type nutrition, which include *C. simula*, coil their proboscis around the victim in a spiral during hunting, retract the proboscis, bringing the prey closer to the mouth opening, and swallow the prey [31]. The current study found that the proboscis significantly contributes to the overall toxicity of the body. Additionally, more than half of TTXs in the *C.* cf. *simula* body were found in the anterior region. According to the data obtained, two possible mechanisms regarding the use of toxins by nemerteans during hunting are suggested: (1) TTX and its analogues, localized in the proboscis, affect the prey, paralyze it, and the nemertean completes the process of killing using TTXs in the foregut; (2) the nemertean kills the victim by TTXs secreted by the front of the body and swallows it without using the proboscis.

Another way animals use TTX is as a defense against predator attacks. It was found that some species of puffer *Takifugu*, in response to stimulation by electric shock, produced secretions containing large amounts of TTX, which was accompanied by inflating the body as a defensive reaction in response to an attack [32]. TTXs are often found in secretions collected from the skin of some animals (e.g., larvae [33], adults of some puffer fish species [34], and flatworms *Planocera multitentaculata* [35]), and according to the authors’ assumptions, these toxins are used to repel predators. TTX was also found in *C. simula* (named *C. linearis*) secretions produced in response to repeated mechanical stimulation for six days [3]. The toxicity of the secretions decreased on the third day and completely disappeared by the last day. In the current study, *C.* cf. *simula* secreted from 667.3 to 2067.6 ng of TTXs, which corresponds to approximately 3.8 and 11.6 MU, and did not exceed 1.4% of the sum of all toxins contained in a specimen. Thus, we assumed the amounts of excreted TTX were sufficient to repel and/or kill potential predators, and the decrease in TTX concentrations in the secretions of *C. simula* (named *C. linearis*), observed by Ali et al. [3], could be explained by the low rate of TTX recovery in the skin of nemerteans upon repeated stimulation. However, this assumption needs to be verified. Nevertheless, Kajihara et al. [15] showed that *C. simula* could be consumed by puffer fish *Takifugu niphobles* resistant to TTX and could be one of the potential sources of the toxin for puffers.

## 4. Conclusions

In the current study, the profile and concentration of TTX and its analogues in the intestine, body wall, and secretions of the nemertean *Cephalothrix* cf. *simula* were examined. As a result, the possible ways of TTX adsorption and the functions of toxins in nemerteans were assumed. Food objects are suggested as the source of TTXs in nemerteans, as evidenced by the high concentration of TTXs in the anterior region of the digestive tract of *C.* cf. *simula* (up to 85% of the body TTXs), probably due to the higher release of toxins from the food object as a result of lysis. At the same time, identical qualitative toxin compositions in the intestine and the body wall may indicate the uptake of TTX analogues with food, not as a result of TTX conversion. The ratio of toxins between different regions of the nemertean intestine varied. However, the topic of whether unequal analogues localization is associated with different absorptions of toxins, due to their physicochemical properties or with the possible existence of an additional TTXs transfer mechanism that selectively acts towards certain analogues, remains to be studied. The significant contribution of the proboscis and anterior intestine to the overall TTXs of *C*. cf. *simula* may indicate that the toxins are utilized by nemerteans in hunting and consuming prey. It was also revealed that the amount of TTXs released by *C.* cf. *simula* in secretions did not exceed 1.4%, but that amount was sufficient to repel and/or kill potential predators.

TTXs were found in a number of nemertean species; however, only several *Cephalothrix* species accumulate it in high enough amounts to be considered “extremely toxic” animals. This feature of nemerteans from the genus *Cephalothrix* may result from a unique source of toxins, which, most likely, is food objects; however, the role of its own microbiota in TTXs production cannot be neglected [36]. A promising approach to uncover the origin of “extremely toxic” species is to identify their food items using metagenomic analysis, followed by screening the identified objects for the presence of TTX and its analogues, as well as by studying their microbiota [37]. This complex approach can help to identify environmental pathways of TTX and its analogues in the ecosystem.

## 5. Materials and Methods

### 5.1. Sample Collection

*Cephalothrix* cf. *simula* nemerteans were collected from the rhizoids of the brown alga *Saccharina* sp. in Spokoynaya Bay (42.7090 N, 133.1809 E), Peter the Great Gulf (Sea of Japan) in July–August 2020. Nemertean species were identified by Dr. Alexey V. Chernyshev, an expert on nemertean zoology. The worms belonged to the *Cephalothrix simula* s. l. group and appeared under the name *Cephalothrix simula* in our early publications. However, recent genetic analysis showed that individuals from the Peter the Great Gulf are not true *C. simula* [15] and should be cited as *C.* cf. *simula*. The worms were kept in seawater aquaria at 17 °C. Some worms were used to collect secretions; for this, nemerteans were placed on a Petri dish with sterile seawater (1 mL) and were exposed to a short (2 s) 6-volt electric current pulse. The voltage was chosen to ensure that nemerteans remained active after stimulation. The secretions were collected in 15 mL tubes, and the resulting samples were frozen at −20 °C for further study. After stimulation, samples of worm bodies were frozen at −20 °C for further study. A number of *C.* cf. *simula* nemerteans were used to study TTXs localization; for this, the proboscis was separated from each individual, the body was divided into three equal parts, the body wall and intestines were separated from each part, and all samples were frozen individually at −20 °C for further research.

### 5.2. TTX and Its Analogues Extraction from the Cephalothrix cf. simula Body and Secretions

The extracts of *C*. cf. *simula* were prepared using the following procedure. Samples were homogenized in 0.1% solution of acetic acid in 70% methanol (1:10 ratio (*v*/*v*) for the body, and 1:3 ratio (*v*/*v*) for the secretions extraction) for 5 min using a hand homogenizer. Then, they were ultrasonically treated with an ultrasonic homogenizer HD 2070 (Bandelin Sonopuls, Berlin, Germany) for 10 min (frequency—20 kHz, amplitude—228 µm, working cycle—0.8 s, and interval—0.2 s). The homogenates were centrifuged (14,000× *g*, 10 min, 4 °C), and the supernatants were collected. For body samples, the extraction was repeated twice more (1:2 ratio (*v*/*v*)), and the supernatants were pooled. The body and secretions extracts were evaporated in a rotor evaporator (Labconco, Kansas City, MO, USA) at 60 °C. The resulting precipitates were dissolved in a 0.1% aqueous solution of acetic acid at ratios of 1 mL/g of nemertean tissue and 100 μL/mL of secretion, and they were filtered through Vivaspin turbo centrifugal concentrators (nominal cutoff molecular weight 5 kDa (Sartorius, Goettingen, Germany)). The resulting samples were stored at −20 °C for further research.

The recovery control was made as follows. A solution of 10 ng of TTX in 3 mL of sterile seawater was prepared, and TTX was extracted according the same procedure as described above for the secretions samples. The recovery was 83.6%.

### 5.3. Analysis of TTX and Its Analogues in the Secretions and Body of Cephalothrix cf. simula

TTX and its analogues were identified using HPLC–MS/MS. The HPLC system included two pairs of LC-30 pumps, a SIL-30AC autosampler, a CTO-20A thermostat, an SCL-20A system controller, and a triple quadrupole mass spectrometer LCMS-8060 (ShimadzuEuropa, Duisburg, Germany) with electrostatic spray ionization (ESI). Separation was carried out using a SeQuant ZIC HILIC column (150 × 2.1 mm, 5 μm) (Merck, Darmstadt, Germany) at 40 °C and a flow rate of 0.2 mL/min. A binary gradient was used: mobile phase A, ammonia (5 mM) and formic acid (8 mM) in 94:6 acetonitrile–water; mobile phase B, ammonia (10 mM) and formic acid (20 mM) in water. A gradient profile was used as follows: (a) 0–4.3 min, 15% B; (b) 4.3–16 min, 25% B; (c) 16–20 min, 50% B. The sample volume was 1 μL. A SeQuant ZIC-HILIC guard column (20 × 2.1 mm, 5 μm) (Merck, Darmstadt, Germany) was installed in line before the analytical column through a two-position 6-port valve. At 4.4 min, the valve was switched and guard columns backflashed with isopropanol (4.4–9 min) and water (9–15 min) at a flow rate of 0.3 mL/min. At 16 min, the valve was switched back. The mass spectrometer was operated in scan (*m*/*z* 200–1000) and multiple reaction monitoring (MRM) modes. The ion source parameters were as follows: interface temperature—380 °C, desolvation line temperature—250 °C, nebulizing gas (N2) flow—3 L/min, drying gas (N2) flow—3 L/min, and heating gas (dry air) flow—17 L/min. Collision energy was 41 eV for precursor transition and 25 eV for fragment transitions. The TTX concentration was calculated using the calibration curve of a standard TTX solution series (Alomone Labs Ltd., Jerusalem, Israel). The toxin detection criteria included a precursor MRM transition peak S/N ratio >3, relative intensity of the fragment ion peak > 4%, and the order of toxins elution corresponded to that described in Bane et al. [38]. MRM transitions were used to detect TTX analogues as described by Vale [39], Kudo et al. [40], Bane et al. [14], Puilingi et al. [41], and Turner et al. [42]. The concentrations of TTX analogues were calculated following the procedure of Chen et al. [43]. The method was validated using standard TTX solutions in MRM mode. The linearity range was from 0.6 to 100 ng/mL. The recovery range from 1 to 100 ng/mL of TTX was 98.4%. The LoQ was 0.6 ng/mL. The limit of detection (LoD) was 0.2 ng/mL, and the relative SD was 4.5%–14.6%.

## Figures and Tables

**Figure 1 toxins-12-00745-f001:**
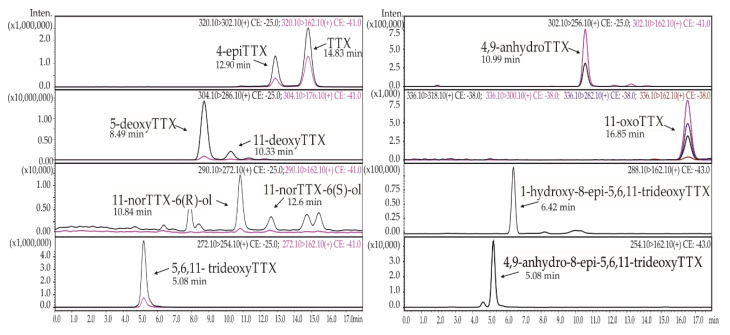
Representative high-performance liquid chromatography–tandem mass spectrometry (HPLC–MS/MS) chromatograms of tetrodotoxin (TTX) and its analogues from *Cephalothrix* cf. *simula* extracts obtained from two experiments.

**Figure 2 toxins-12-00745-f002:**
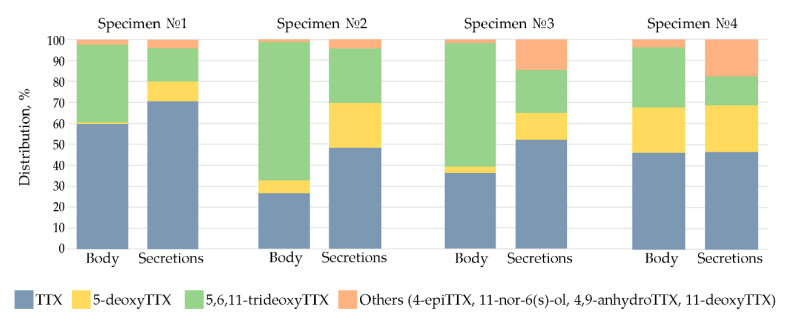
Proportion of tetrodotoxin (TTX) and its analogues in the body and secretions of *Cephalothrix* cf. *simula*.

**Figure 3 toxins-12-00745-f003:**
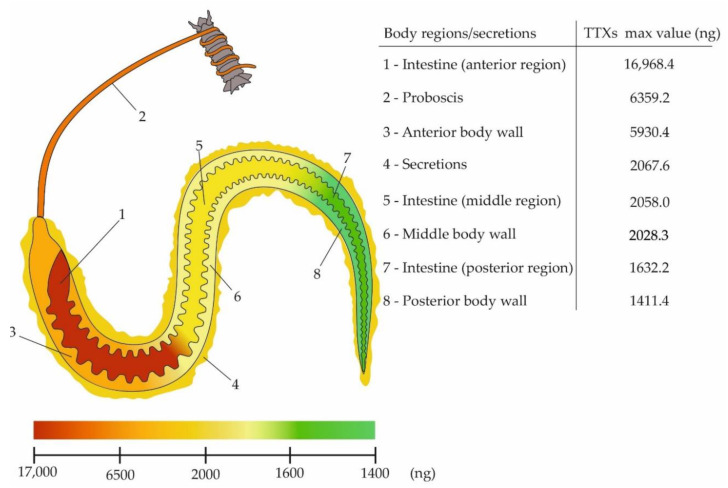
Schematic illustration of the maximum levels (ng) of tetrodotoxin and its analogues (TTXs) in *Cephalothrix* cf. *simula*.

**Table 1 toxins-12-00745-t001:** Tetrodotoxin (TTX) and its analogues of *Cephalothrix* cf. *simula* in the body and secretions.

No. of Specimen	Weight (g)	Sample	TTX and Its Analogues (TTXs) (ng)								
			TTX	4-epiTTX	11-norTTX-6(s)-ol	4,9-anhydroTTX	11-deoxyTTX	5-deoxyTTX	5,6,11-trideoxyTTX	Sum of TTXs	Total Amount of TTXs
1	0.20	body	254,758.0	10,641.2	13.2	282.0	338.5	3340.2	158,354.4	427,727.5	428,596.9
		secretions	613.0	29.7	ND	4.2	1.0	83.2	138.3	869.4	
2	0.17	body	51,323.0	2137.1	6.2	91.9	94.0	11,573.8	127,065.6	192,291.6	193,059.9
		secretions	369.2	25.0	2.2	5.0	1.0	165.7	200.2	768.3	
3	0.25	body	53,233.0	1500.4	ND	329.1	542.4	4311.5	87,096.5	147,013.0	149,080.5
		secretions	1076.1	240.2	ND	58.7	5.1	265.0	422.5	2067.6	
4	0.30	body	45,345.0	1813.3	ND	467.5	1334.7	20,870.9	28,268.8	98,100.2	98,767.5
		secretions	308.6	69.6	ND	44.0	4.5	148.5	92.1	667.3	

ND: not detected.

**Table 2 toxins-12-00745-t002:** Tetrodotoxin (TTX) and its analogues in different regions of *Cephalothrix* cf. *simula*.

No. of Specimen	Weight (g)	Organ	Region	TTX and Its Analogues (TTXs) (ng)													
				TTX	4-epiTTX	11-norTTX-6(s)-ol	11-norTTX-6(r)-ol	4,9-anhydroTTX	11-deoxyTTX	5-deoxyTTX	5,6,11-trideoxyTTX	11-oxoTTX	4,9-anhydro-8-epi-5,6,11-trideoxyTTX	1-hydroxy-8-epi-5,6,11-trideoxyTTX	Sum of TTXs	Sum of TTXs in the Organ	Total Amount of TTXs
5	0.17	Intestine	Anterior	9608.3	539.8	9.4	42.8	47.4	86.6	3131.7	3487.5	<LoQ	9.9	5.0	16,968.4	19,954.5	29,012.6
			Middle	561.3	53.6	0.7	4.6	5.9	37.1	380.6	308.5	<LoQ	1.0	0.6	1354.9		
			Posterior	582.8	93.0	0.7	2.4	8.6	6.1	621.7	315.6	<LoQ	0.6	0.7	1632.2		
		Body wall	Anterior	714.5	42.5	0.7	3.5	4.0	6.6	257.5	204.1	<LoQ	0.9	0.4	1234.7	2698.9	
			Middle	491.9	44.0	0.4	3.2	4.7	12.4	337.4	239.5	<LoQ	0.9	0.6	1135.0		
			Posterior	122.3	15.5	0.1	0.3	1.9	1.7	138.5	48.3	<LoQ	0.3	0.3	329.2		
		Proboscis		2534.6	493.4	48.2	31.4	35.9	34.9	1992.4	1177.9	2.0	4.3	4.2	6359.2	6359.2	
6	0.07	Intestine	Anterior	1342.1	15.2	0.4	0.8	3.2	12.5	194.1	627.3	ND	1.5	0.2	2197.3	4395.2	15,190.3
			Middle	873.9	18.4	ND	0.7	1.6	9.6	142.8	563.9	ND	1.1	0.3	1612.3		
			Posterior	301.8	9.1	0.1	0.3	1.0	9.0	83.8	179.9	ND	0.4	0.2	585.6		
		Body wall	Anterior	4134.5	47.7	1.0	ND	6.4	6.0	358.3	1372.4	ND	4.1	ND	5930.4	9370.1	
			Middle	1112.0	10.7	0.2	0.8	2.0	3.0	173.1	724.0	ND	2.2	0.3	2028.3		
			Posterior	630.7	10.0	ND	0.7	2.3	11.6	189.0	564.8	ND	1.5	0.8	1411.4		
		Proboscis		900.0	9.4	2.9	1.5	1.6	3.2	115.2	389.8	ND	1.2	0.2	1425.0	1425.0	
7	0.10	Intestine	Anterior	2836.2	42.0	1.1	3.7	9.6	19.5	1013.6	978.6	ND	3.2	0.5	4908.0	7453.9	13,065.2
			Middle	1038.7	19.4	0.8	1.7	6.0	9.2	490.9	489.2	ND	1.7	0.4	2058.0		
			Posterior	186.7	8.5	0.1	0.5	1.8	2.0	175.7	111.8	ND	0.5	0.3	487.9		
		Body wall	Anterior	2906.6	24.7	1.8	2.6	5.9	3.5	438.1	463.9	ND	1.8	0.3	3849.2	4730.8	
			Middle	331.2	8.9	0.1	0.4	1.3	0.5	131.1	122.9	ND	0.5	0.1	597.0		
			Posterior	113.5	4.6	0.1	0.3	1.0	0.4	98.2	66.0	ND	0.3	0.2	284.6		
		Proboscis		516.1	7.5	1.9	0.8	1.2	1.3	89.2	261.6	ND	0.9	ND	880.5	880.5	

ND: not detected; LoQ: limit of quantification.

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
