# Peer review of "Tetrodotoxin and Its Analogues in Cephalothrix cf. simula (Nemertea: Palaeonemertea) from the Sea of Japan (Peter the Great Gulf): Intrabody Distribution and Secretions"

_toxins, 2020, doi:10.3390/toxins12120745_

Round 1
Reviewer 1 Report
Manuscript ID:Toxins-1020338
Type of manuscript :Article
General Comments for Authors can be seen in the attached file.

Author Response
We wish to express our appreciation to the reviewer for the valuable comments and for the overall positive feedback for our paper. Our responses to the reviewer’s comments are provided below. All changed and added text in the manuscript is highlighted with a yellow color.
Responses to Reviewer 1
1. L152
Change 25.590 MU/g into 25,590 MU/g.
Response: Done, line 144.
2. L304
5.1 Sample collection
The nemertean individuals of Cephalothrix cf. simula were collected from the rhizoids of the brown alga Saccharina sp. in the Spokoynaya Bay, Peter the Great Gulf (Sea of Japan) in October 2019 - August 2020. However, the location for the collection of samples seems to be a little hard for readers to understand. Therefore, it is recommended for you to add map or the information such as latitude/longitude on the location for the collection into this section. Please consider that.
Response: The latitude/longitude of the location was added, line 304.
Reviewer 2 Report
The manuscript identified the distribution of tetrodotoxin (TTX) and its analogs in different regions and organs of Cephalothrix cf. simula, showing that 1) the same qualitative composition of TTXs throughout the body but differences in the total concentration of the toxins and 2) anterior region of the body contains the highest TTXs. Based on the findings, the authors provide the possible way of TTX adsorption and functions of the TTXs in nemerteans.
Overall, this is an interesting and scientifically sound work. The manuscript is well written and the scientific content appears to be solid. It might have a good impact in the field.
I have a few comments, as follows.
- For Figure 1, the left column and the third row. Can authors explain what are the peaks around at 8 minute and 14-16 minutes? Are these peaks new TTXs? Also, please label clearly in Figure1 which panels belong to the first and second experiments.
- Line 49, “eleased” should be changed to “released”.
Author Response
We thank the reviewer for the overall positive feedback for our paper. Our responses to the reviewer’s comments are provided below. All changed and added text in the manuscript is highlighted with a yellow color.
Responses to Reviewer
1. For Figure 1, the left column and the third row. Can authors explain what are the peaks around at 8 minute and 14-16 minutes? Are these peaks new TTXs? Also, please label clearly in Figure1 which panels belong to the first and second experiments.
Response: The peaks at 8 minute and 14-16 minutes have relative intensity of the fragment ion peak > 4% and could be referred to co-extracted compounds from the matrix, not to target TTXs. On the Figure 1 one chromatogram was chosen from each of TTXs as representative since peaks of each toxin from different experiments are identical. This information was added in the figure caption, line 64-66: “Figure 1. Representative high-performance liquid chromatography–tandem mass spectrometry (HPLC–MS/MS) chromatograms of tetrodotoxin (TTX) and its analogues from Cephalothrix cf. simula extracts obtained from two experiments.”
2. Line 49, “eleased” should be changed to “released”.
Response: Done, line 49.